# HIF1A-mediated pathways promote euploid cell survival in chromosomally mosaic embryos

Estefania Sanchez-Vasquez, Marianne E Bronner, Magdalena Zernicka-Goetz*

Division of Biology 139-74, California Institute of Technology, Pasadena, United States

## eLife Assessment

Sanchez-Vasquez et al establish an innovative approach to induce aneuploidy in preimplantation embryos. This **important** study extends the author's previous publications evaluating the consequences of aneuploidy in the mammalian embryo. In this work, the authors investigate the developmental potential of aneuploid embryos and characterize changes in gene expression profiles under normoxic and hypoxic culture conditions. Using a **solid** methodology they identify sensitivity to Hif1alpha loss in aneuploid embryos, and in further **convincing** experiments they assess how levels of DNA damage and DNA repair are altered under hypoxic and normoxic conditions.

*For correspondence:
magdaz@caltech.edu

Competing interest: The authors declare that no competing interests exist.

**Abstract** Human fertility is suboptimal in part by error-prone divisions during early cleavage stages, which frequently result in chromosomal aneuploidy. Most human pre-implantation embryos are mosaics of euploid and aneuploid cells, yet those with a low proportion of aneuploid cells can develop to term at rates similar to fully euploid embryos. How embryos manage aneuploidy during early development remains poorly understood – yet this knowledge is crucial for improving fertility outcomes and reducing developmental defects. To investigate these mechanisms, we established a new mouse model of chromosome mosaicism to trace the fate of aneuploid cells during pre-implantation development. We previously used the Mps1 inhibitor reversine to induce aneuploidy. Here, we demonstrate that the more specific Mps1 inhibitor AZ3146 similarly disrupts chromosome segregation but supports higher developmental potential than reversine. AZ3146-treated embryos transiently upregulate hypoxia-inducible factor-1A (HIF1A) without triggering *Trp53* activation. Given that pre-implantation embryos develop in a hypoxic environment in vivo, we further explored the role of oxygen tension. Hypoxia exposure in vitro reduced DNA damage in response to Mps1 inhibition and increased the proportion of euploid cells in mosaic epiblast. Conversely, HIF1A inhibition decreased the proportion of aneuploid cells. Together, these findings uncover a role for hypoxia signaling in modulating the response to chromosomal errors and suggest new strategies to improve the developmental potential of mosaic human embryos.

## Introduction

Humans have relatively low fertility compared to other mammals, with only ~30% of conceptions resulting in live birth (*Palmerola et al., 2022*; *Capalbo et al., 2021*). A striking feature of early human development is the high frequency of chromosome segregation errors during cleavage stage divisions, leading to aneuploidy – the gain or loss of chromosomes (*Allais and FitzHarris, 2022*; *Currie et al., 2022*; *Pauerova et al., 2020*; *Vanneste et al., 2009*). This high incidence of aneuploidy is thought to underlie low human fecundity and many developmental defects (*Palmerola et al., 2022*). Both in vivo

and in vitro fertilization (IVF) frequently result in embryos that are chromosomally mosaic, containing a mixture of diploid and aneuploid cells. It is estimated that ~60% of pre-implantation IVF embryos exhibit this form of mosaicism (*Capalbo et al., 2021*). Despite its prevalence, our understanding of how embryos respond to and cope with aneuploidy during early development remains limited.

The incidence of aneuploidy declines as development progresses (*Shahbazi et al., 2020*; *Yang et al., 2021*), but the mechanism underlying this decline remains unclear. Remarkably, mosaic human embryos can develop to term (*Capalbo et al., 2021*; *Greco et al., 2015*; *Starostik et al., 2020*). Specifically, embryos classified as low- or medium-grade mosaics – defined by the presence of 20–30% or 30–50% aneuploid cells in the trophectoderm (TE) – have implantation and birth rates comparable to fully euploid embryos (*Capalbo et al., 2021*). These observations raise a fundamental question: How do some mosaic embryos maintain developmental potential despite chromosomal abnormalities? Uncovering the mechanisms that support the viability of mosaic embryos is essential for improving reproductive outcomes and embryo selection strategies.

Mouse models of chromosome mosaicism provide a powerful system to investigate mechanisms that cannot be ethically studied in human embryos. Importantly, mouse and human pre-implantation development are highly similar: both undertake cleavage divisions, compaction, blastocyst cavity formation, and hatching, albeit with slightly different timings (*Molè et al., 2020*; *Zhu et al., 2021*). During this period, outer cells differentiate into the extra-embryonic TE, whereas inner cells form the inner cell mass (ICM), which further segregates into the embryonic epiblast (EPI) or extra-embryonic primitive endoderm (PE). The TE will form the placenta, the PE will form the yolk sac, and the EPI will form the fetus (*Zhu and Zernicka-Goetz, 2020*).

Recently, our group developed the first mouse model of chromosome mosaicism by inducing aneuploidy using the small-molecule inhibitor reversine (*Bolton et al., 2016*; *Singla et al., 2020*). Reversine is a pan-Aurora kinase inhibitor that also antagonizes the A3 adenosine receptor and inhibits mitotic kinase monopolar spindle 1 (MPS1) (*D'Alise et al., 2008*; *Santaguida et al., 2010*). We found that reversine-treated mosaic embryos exhibit widespread aneuploidy and upregulation of p53 (*Bolton et al., 2016*). Importantly, aneuploid cells in mosaic embryos are progressively eliminated from the EPI, beginning around the time of implantation. Consistent with findings in human embryos (*Capalbo et al., 2021*), murine mosaic embryos containing at least 50% of euploid cells had a similar developmental potential to fully euploid embryos (*Bolton et al., 2016*; *Singla et al., 2020*).

However, because reversine affects p53 and may compromise cellular fitness (*D'Alise et al., 2008*; *Santaguida et al., 2010*), we sought to establish a complementary model using a more specific Mps1 inhibitor, AZ3146 (*Hewitt et al., 2010*). Although both AZ3146 and reversine interfere with the spindle assembly checkpoint, they bind to Mps1 in a different manner (*Lan and Cleveland, 2010*). In mouse embryos, AZ3146 treatment was shown to double the occurrence of micronuclei, indicative of chromosome segregation defects, but without impairing overall cellular fitness (*Vázquez-Diez et al., 2019*). In this study, we use both AZ3146- and reversine-treated embryos to dissect the mechanisms governing aneuploid cell elimination and survival during pre-implantation development.

## Results

### AZ3146 treatment induces chromosome segregation defects in pre-implantation embryos

To generate distinct models of aneuploidy, we treated 4- to 8-cell stage mouse embryos with AZ3146 (20 μM) (*Vázquez-Diez et al., 2019*), as well as with reversine (0.5 μM) (*Bolton et al., 2016*) as a positive control or DMSO (vehicle) as a negative control (*Figure 1A*). We evaluated how the different Mps1 inhibitors affect chromosome segregation by detecting nuclei and kinetochores in 8-cell embryos and counting chromosomes in situ (*Pauerova et al., 2020*). Micronuclei were identified as small DAPI-stained chromosomes that were clearly distinct from the nuclei (*Figure 1—figure supplement 1A*). We examined 32 DMSO-treated control embryos and observed only 22 cells with micronuclei out of a total of 256 cells (*Figure 1—figure supplement 1B*). In contrast, we observed 82 cells with micronuclei in a total of 144 individual cells from 18 reversine-treated embryos, and 182 cells in a total of 304 individual cells from 38 AZ3146-treated embryos (*Figure 1—figure supplement 1A*). After normalization, the average of aneuploidy in DMSO-treated 8-cell embryos was 12.5%, compared to 75% in the reversine-treated embryos and 62.5% in the AZ3146-treated embryos (*Figure 1—figure supplement*

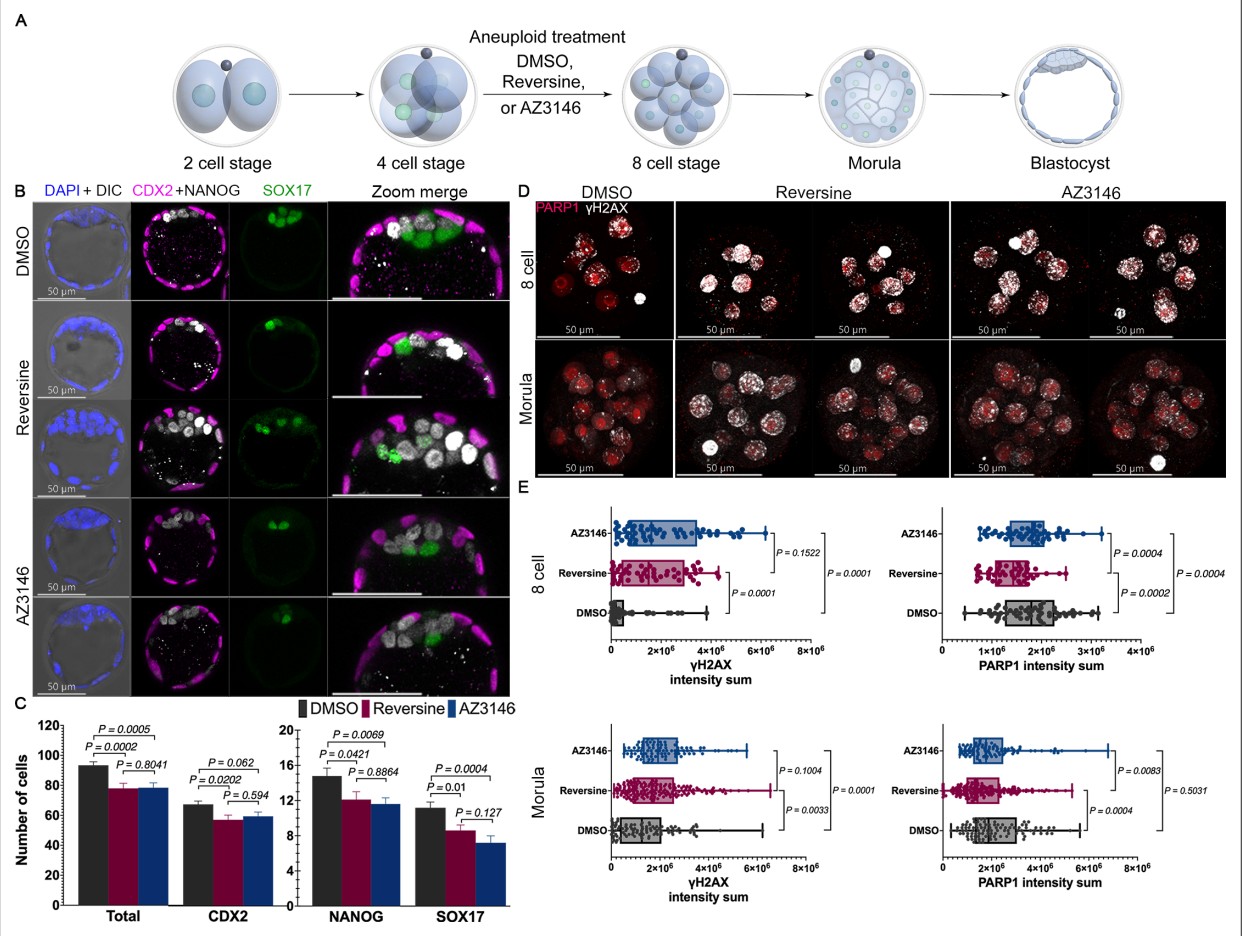

**Figure 1.** Lineage analysis of aneuploid embryos generated by selective Mps1 inhibitors AZ3146 and reversine. (**A**) Graphic representation of 4-cell embryos treated with DMSO (control) or Mps1 inhibitors reversine (0.5 μM) and AZ3146 (20 μM) to inactivate the spindle assembly checkpoint (SAC) and induce chromosome segregation errors. After washing, embryos were cultured to the mature blastocyst stage (E4.5) and analyzed for lineage specification. (**B**) Immunofluorescence imaging of well-known lineage markers CDX2 (trophectoderm [TE]), NANOG (epiblast [EPI]), and SOX17 (primitive endoderm [PE]) reveals that overall embryonic morphology and cavitation is not affected by Mps1 inhibition. (**C**) Number of cells in each lineage was quantified to evaluate the effect of drug treatment on blastocyst development. Importantly, both reversine and AZ3146 treatments reduce the number of cells in the inner cell mass (ICM), marked by NANOG (EPI) and SOX17 (PE). Whereas the TE, marked by CDX2, is reduced only in the reversine-treated embryos (n=28 embryos per treatment, cumulative of three independent experiments. Statistical test: Mann–Whitney U-test, error bars represent s.e.m.). (**D**) Analysis of DNA damage and DNA repair based on immunofluorescence against γH2A.X (phosphorylated form of H2A.X, white) and PARP1 (red), respectively. (**E**) Intensity analysis shows that reversine and AZ3146 increase DNA damage at the 8-cell stage through morula stage compared with DMSO-treated embryos. Importantly, reversine appears to downregulate PARP1 expression at the 8-cell stage, which extends to morula stage embryos (n=~15 embryos per treatment, ~120 cells. Collection was made during three independent experiments. Statistical test: t-test, error bars represent s.e.m.).

The online version of this article includes the following source data and figure supplement(s) for figure 1:

**Figure supplement 1.** AZ3146-treated embryos are aneuploid and can develop to post-implantation stages.

**Figure supplement 1—source data 1.** Aneuploidy quantification in embryos after in situ chromosome counting.

**Figure supplement 2.** PARP1 is required for inner cell mass (ICM) development.

*1B*). We also detected nondividing nuclei as rounded circles with distinct DAPI intensities in embryos treated with reversine and AZ3146. Specifically, we observed 28 nondividing cells (19%) in reversine-treated embryos and 24 nondividing cells (7.9%) in AZ3146-treated blastomeres, but none in DMSO-treated blastomeres (*Figure 1—figure supplement 1B*). The increased frequencies of micronuclei and non-dividing cells in response to Mps1 inhibition are consistent with elevated aneuploidy and chromosomal instability (*Daughtry and Chavez, 2016*; *Vázquez-Diez and FitzHarris, 2018*), as we showed previously for reversine-treated embryos (*Bolton et al., 2016*).

Given these distinct responses to reversine and AZ3146, we examined how these treatments impact the lineages in the blastocyst. To this end, we treated embryos with DMSO, reversine, or AZ3146 from the 4- to 8-cell stage, and then washed and cultured them until the late blastocyst stage (E4.5). We assessed lineages by immunofluorescence (IF) to detect the TE marker CDX2, the EPI marker NANOG, and the PE marker SOX17. We found that all lineages segregated normally, and that blastocyst morphology was similar in all three conditions (*Figure 1B*). We quantified the number of cells in the blastocysts and found that DMSO-treated controls had a median of 93 cells in total, whereas reversine- and AZ3146-treated embryos had a median of only 79 and 82 cells, respectively (n=28 embryos per treatment collected from three independent experiments; *Figure 1C*). DMSO-treated blastocysts contained a median of 15 EPI, 11 PE, and 67 TE cells, whereas reversine-treated blastocysts contained 12 EPI, 8 PE, and 59 TE cells, and AZ3146-treated blastocysts had 11 EPI, 7 PE, and 64 TE cells (*Figure 1C*). These data suggest that reversine treatment compromises the development of all three lineages, as observed previously (*Bolton et al., 2016*), whereas AZ3146 mainly compromises ICM (EPI and PE) development (statistical test: Mann–Whitney U-test).

To compare the developmental potential of blastocysts that had been treated with AZ3146 or reversine, we transfer ~7 embryos per treatment into opposite uterine horns of the same mouse and counted decidua, which reflect successful implantation, as well as viable embryos at E9.5. Although two (8%) reversine-treated blastocysts developed decidua (*Figure 1—figure supplement 1C*), none gave rise to viable E9.5 embryos. In contrast, seven (30%) AZ3146-treated blastocysts developed decidua (*Figure 1—figure supplement 1C*) and five (21%) generated viable E9.5 embryos (*Figure 1—figure supplement 1D*). Thus, AZ3146-treated embryos appear to have a higher developmental potential than reversine-treated embryos.

Considering that cellular fitness and aneuploid stress are related to DNA damage and DNA repair, we first sought to understand if these parameters were affected in our drug treatments. We used IF to quantify the levels of the DNA repair marker PARP1 and of the DNA damage marker, phosphorylated H2A.X ($\gamma$H2A.X) at the 8-cell and morula stages (*Figure 1D*). We observed elevated $\gamma$H2A.X levels in reversine- and AZ3146-treated 8-cell embryos compared to controls (*Figure 1E*) (n=~15 embryos per treatment, ~120 cells. Collection was made during three independent experiments. Statistical test: t-test), which eventually returned to normal in AZ3146-treated blastocysts, but not in reversine-treated blastocysts (*Figure 1—figure supplement 2C*).

PARP1 was specifically enriched in the EPI lineage in normal blastocysts, as assessed by IF (n>21 embryos per treatment collected from three independent experiments. Statistical test: Mann–Whitney U-test) and re-analysis of published scRNA-seq data (*Deng et al., 2014*; GEO accession: GSE45719) (*Figure 1—figure supplement 2B*). Intriguingly, PARP1 levels were reduced at the 8-cell and morula stages in reversine-treated embryos compared to controls and to AZ3146-treated embryos (*Figure 1D and E*). Moreover, late morula stage embryos treated with the PARP inhibitor olaparib (10 µM) (*Hou et al., 2022*; *Prasad et al., 2017*) developed into smaller blastocysts with only 78 total cells, reflecting a relatively low number of ICM cells, only 8 EPI and 4 PE cells in DMSO-treated embryos (*Figure 1—figure supplement 2C–E*). Treatment with reversine, but not AZ3146, further reduced the number of cells in the EPI lineage of olaparib-treated embryos (*Figure 1—figure supplement 2E*) (n=15 embryos per treatment collected from three independent experiments. Statistical test: Mann–Whitney U-test). Overall, these data suggest that PARP1 is required for proper development of the ICM, and that its reduced levels after reversine treatment may negatively impact development.

## Reversine and AZ3146 activate distinct stress response pathways in pre-implantation embryos

Chromosome mis-segregation and aneuploidy are associated with different cellular stress pathways (*Zhu et al., 2018*). For instance, p53 is frequently activated following DNA damage, which can limit proliferation and trigger apoptosis (*Abuetabh et al., 2022*; *Soto et al., 2017*; *Thompson and Compton, 2010*). The p38 mitogen-activated protein kinase can also be activated following DNA damage (*Thompson and Compton, 2010*). In aneuploid cells, p38 promotes apoptosis by inhibiting the transcription factor *Hif-1a* (*Simões-Sousa et al., 2018*), which otherwise promotes cell survival, proliferation, and metabolic changes in aneuploid cells and in response to hypoxia in different contexts (*Hu et al., 2003*; *Simões-Sousa et al., 2018*).

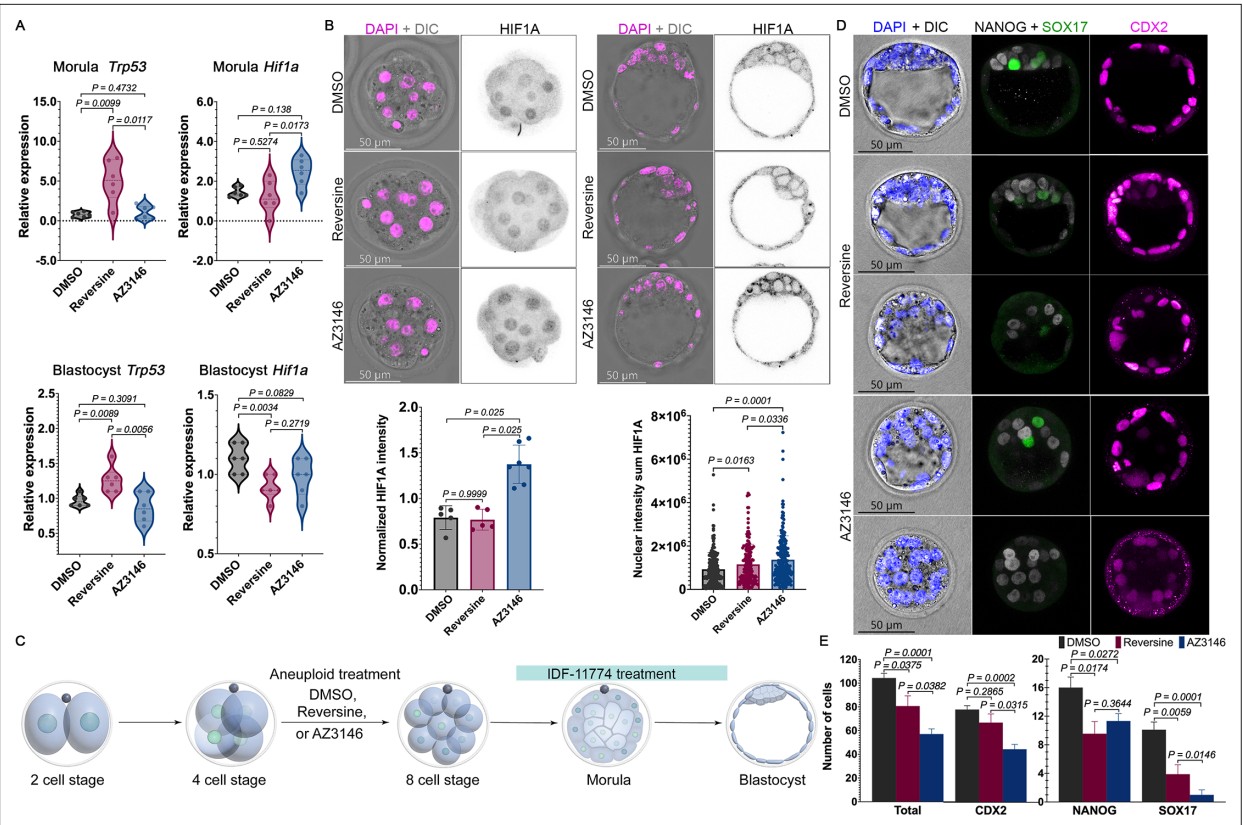

**Figure 2.** AZ3146-treated embryos elevate HIF1A activity to support formation of the trophectoderm (TE) and primitive endoderm (PE). (**A**) qPCR analysis of *Trp53* and *Hif1a* mRNA expression at morula and blastocyst stages reveals that *Trp53* is upregulated in reversine-treated embryos and that *Hif1a* is upregulated in AZ3146-treated embryos at morula stages (sample: three biological replicates and two technical replicates per experiment with each replicate having a minimum of 16 embryos. Statistical test: Welch's t-test). (**B**) Immunofluorescence against HIF1A (black) shows an increase in nuclear intensity in AZ3146-treated embryos at morula stages. At the blastocyst stage, nuclear and cytoplasmic HIF1A are increased in AZ3146-treated embryos; normalization was based on DAPI staining (magenta) (n=~20 embryos per treatment. Statistical test: Mann–Whitney U-test, error bars represent s.e.m.). (**C**) Graphic representation of 4-cell embryos treated with DMSO and aneuploid drugs. Downregulation of HIF1A was achieved by treatment with IDF-11774 immediately after wash of the aneuploid drugs. Immunofluorescence analysis of lineage specification in blastocyst cultured with the HIF1A inhibitor IDF-11774 using antibodies against CDX2 (TE), NANOG (EPI), and SOX17 (PE). Importantly, IDF-11774 appears to affect cavitation of some AZ3146-treated embryos. (**D**) Lineage analysis at the blastocyst stage shows that TE and PE specification are affected by IDF-11774 treatment. The number of cells in each lineage was quantified to evaluate the effect on blastocyst development (n=20 embryos per treatment collected from three independent experiments. Statistical test: Mann–Whitney U-test, error bars represent s.e.m.).

The online version of this article includes the following source data and figure supplement(s) for figure 2:

**Source data 1.** qPCR quantifications for Trp53 and Hif1a in morulas and blastocysts.

**Figure supplement 1.** Pharmacological inhibitors of HIF1A have distinct effects on mouse pre-implantation embryos.

To investigate how treatment with reversine or AZ3146 affects *Trp53* and *Hif1a* expression in mouse embryos, we performed RT-qPCR at the morula and blastocyst stages. We normalized to *Ppia* (peptidylprolyl isomerase A) mRNA, which is a stable reference gene in diploid and polyploid embryos (*Gu et al., 2014*). Reversine-treated embryos displayed a significant increase in *Trp53* transcript levels at the morula (7-fold) and blastocyst (1.3-fold) stages compared with DMSO-treated embryos (sample: three biological replicates and two technical replicates per experiment with each replicate having a minimum of 16 embryos. Statistical test: Welch's t-test) (*Figure 2A*), as we showed previously (*Singla et al., 2020*). In addition, reversine-treated embryos showed reduced expression of *Hif1a* at the blastocyst stage, which would be consistent with p38 activation. In contrast, AZ3146-treated embryos did not show upregulation of *Trp53* at either the morula or blastocyst stage (*Figure 2A*). Moreover, AZ3146-treated embryos showed a transient increase in *Hif1a* mRNA levels (threefold) at the morula stage compared to DMSO- and reversine-treated embryos (*Figure 2A*), which returned to normal levels at the blastocyst stage. HIF1A protein was present in DMSO- and reversine-treated embryos

and elevated at the morula and blastocyst stages in AZ3146-treated embryos (n=~20 embryos per treatment. Statistical test: Mann–Whitney U-test, error bars represent s.e.m.) (*Figure 2B*). HIF1A appeared to be mostly nuclear in morula, but mostly cytoplasmic in blastocysts, under all three conditions (*Figure 2B*). Overall, these data suggest that treatment with reversine, but not AZ3146, induces multiple stress pathways in pre-implantation embryos. These differences may contribute to the increased developmental potential of embryos treated with AZ3146 versus reversine.

## HIF1A activity is required for proper blastocyst formation after Mps1 inhibition

It was previously shown that *Hif1a*$^{-/-}$ embryos undergo developmental arrest and lethality by E11 (*Iyer et al., 1998*). To assess the role of HIF1A in the embryo's response to reversine and AZ3146, we used a pharmacological approach to inhibit its function. Briefly, we tested two different small molecules that have been shown to inhibit HIF1A activity, PX-478 (*Lee and Kim, 2011*; *Zhao et al., 2015*) and IDF-11774 (*Ban et al., 2017*). We treated control zygotes with DMSO, PX-478 (2 μM), or IDF-11774 (20 μM) until the blastocyst stage (*Figure 2—figure supplement 1A*). Unlike PX-478, IDF-11774 treatment did not significantly affect the total number of cells in the TE or the whole blastocyst compared to the control (n=~12 embryos per treatment collected from three independent experiments. Statistical test: Mann–Whitney U-test) (*Figure 2—figure supplement 1B*). Therefore, we used IDF-11774 to inhibit HIF1A in subsequent experiments.

Next, we treated embryos with DMSO, reversine, or AZ3146 from the 4- to 8-cell stage, washed them, and then inhibited HIF1A with IDF-11774 from the 8-cell to blastocyst stage (*Figure 2C*). Inhibition of HIF1A did not abolish cavitation but dramatically lowered the number of TE and especially PE cells in AZ3146-treated embryos compared to DMSO-treated controls (*Figure 2D*) and compared to AZ3146-treated embryos not exposed to IDF-11884 (*Figure 1C*). Inhibition of HIF1A also lowered the number of cells in reversine-treated embryos compared to controls, but the effect was smaller (n=20 embryos per treatment collected from three independent experiments. Statistical test: Mann–Whitney U-test) (*Figure 2D*). Overall, our data suggest that elevated HIF1A activity from the 8-cell stage onward is particularly important to promote the survival of aneuploid TE and PE cells after AZ3146 treatment.

Taken together, our data suggest that AZ3146 and reversine have distinct effects on the pre-implantation embryo. Compared to reversine-treated embryos, AZ3146-treated embryos appear to have increased developmental potential, lack upregulation of *Trp53*, and show transient upregulation of *Hif1a*. Moreover, AZ3146-treated embryos have a greater dependence on HIF1A activity to form the TE and PE.

## Hypoxia exposure attenuates DNA damage and blastomere defects in response to Mps1 inhibition

Pre-implantation development occurs in a hypoxic environment, and many clinics culture human embryos under hypoxic conditions (5%) (*Houghton, 2021*). It was reported that physiologic oxygen concentration (~5% oxygen) can improve the yield and quality of mammalian blastocysts (*Ciray et al., 2009*; *Nguyen et al., 2020*) and increase the nuclear translocation of HIF1A in mouse blastocysts (*Choi et al., 2021*). To investigate how hypoxia affects the development of euploid and aneuploid mouse embryos, we repeated our experiments performed under normoxia (20% O$_2$/5% CO$_2$; control) (*Figure 1A*) in hypoxic (5% O$_2$/5% CO$_2$) conditions taking into consideration the standard practices in the IVF clinics (*Knudtson et al., 2022*; *Figure 3A*). We cultured embryos from the 2-cell stage until the blastocyst stage, treating them with DMSO, reversine, or AZ3146 from the 4- to 8-cell stage.

First, we examined lineage specification by quantifying the cell numbers in all three lineages at the blastocyst stage. We found that DMSO-treated blastocysts cultured under hypoxia had a median of 74 cells, representing 14 EPI cells, 6 PE cells, and 54 TE cells (n=18 embryos per treatment collected from three independent experiments. Statistical test: Mann–Whitney U-test) (*Figure 3B and C*). These data suggest that, under our conditions, the cell number in the pre-implantation mouse blastocyst is reduced when cultured in hypoxia compared to normoxia (*Figures 3C and 1C*), particularly in the TE and PE. Notably, reversine and AZ3146 treatment did not further reduce cell numbers for embryos cultured under hypoxia, except for in the PE (*Figure 3C*).

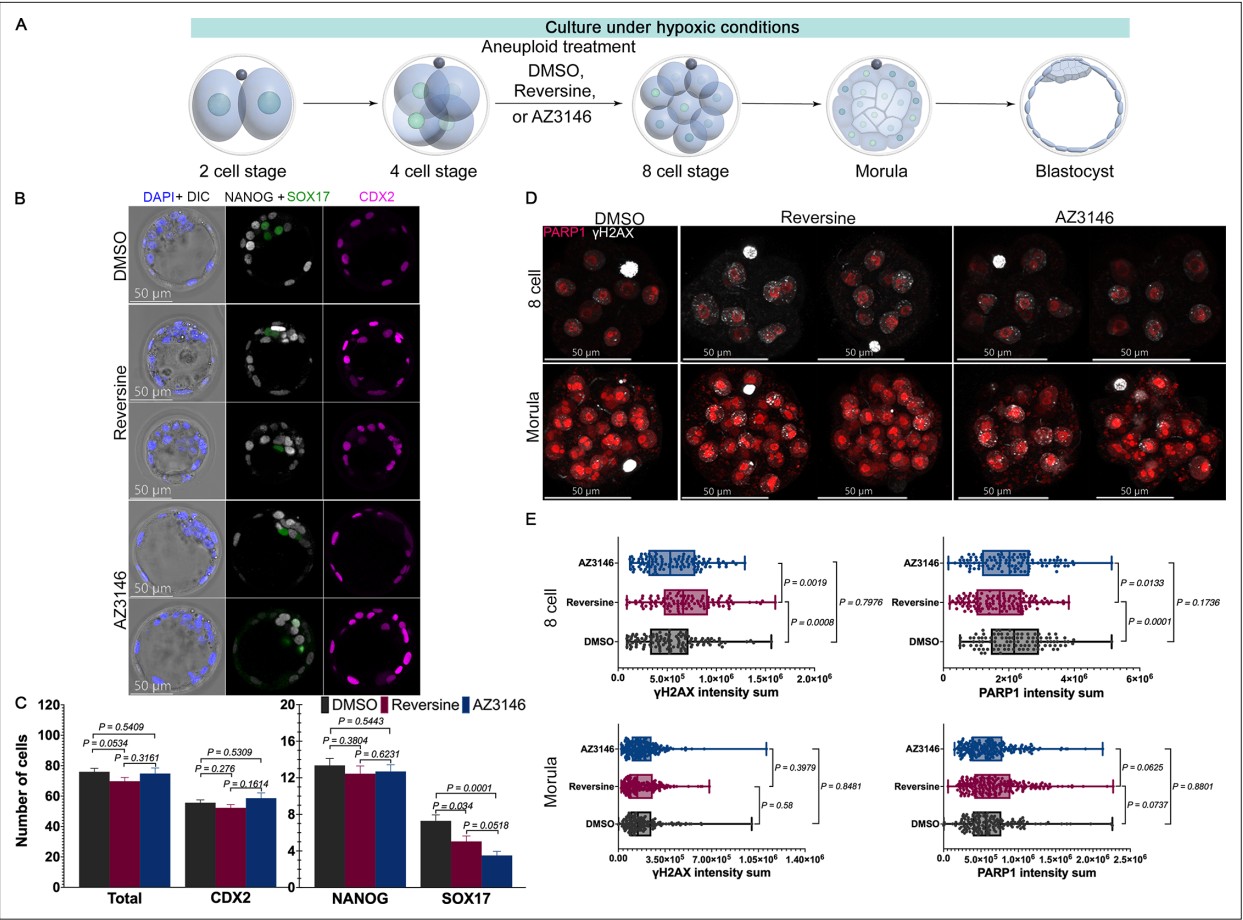

**Figure 3.** Hypoxia exposure reduces DNA damage and affects lineage proportions in the aneuploid blastocyst. (**A**) Graphic representation of the hypoxia experiments. 2-cell embryos were cultured until the blastocyst stage in hypoxia conditions (5% oxygen). As before, 4-cell stage embryos were treated with DMSO or Mps1 inhibitors reversine and AZ3146 until the 8-cell stage. After washing, embryos were cultured to the mature blastocyst stage (E4.5) and analyzed for lineage specification. (**B**) Immunofluorescence imaging of well-known lineage markers CDX2 (trophectoderm [TE]), NANOG (epiblast [EPI]), and SOX17 (primitive endoderm [PE]) reveals that overall embryonic morphology and cavitation is not affected by Mps1 inhibition or hypoxia. (**C**) Lineage analysis at blastocyst stage. The number of cells in each lineage was used to evaluate the effects in blastocyst development (n=18 embryos per treatment collected from three independent experiments. Statistical test: Mann–Whitney U-test, error bars represent s.e.m.). (**D**) Immunofluorescence against PARP1 (red) and γH2A.X (white) in blastocyst after drug treatments. (**E**) Intensity analysis shows that, under hypoxia, at 8-cell stage, DNA damage is increased after exposure to reversine but not AZ3146. PARP1 expression is altered only at the 8-cell stage in reversine-treated embryos (n=25 embryos per treatment collected from three independent experiments. Statistical test: t-test, error bars represent s.e.m.).

The online version of this article includes the following source data for figure 3:

**Source data 1.** Raw fluorescence intensity of PARP1 and DAPI for reversine-treated embryos grown in normoxia.

**Source data 2.** Raw fluorescence intensity of PARP1 and DAPI for control embryos grown in normoxia.

**Source data 3.** Raw fluorescence intensity of PARP1 and DAPI for AZ3146-treated embryos grown in normoxia.

**Source data 4.** Raw fluorescence intensity of PARP1 and DAPI for reversine-treated embryos grown in hypoxia.

**Source data 5.** Raw fluorescence intensity of PARP1 and DAPI for control embryos grown in hypoxia.

**Source data 6.** Raw fluorescence intensity of PARP1 and DAPI for AZ3146-treated embryos grown in hypoxia.

To investigate how hypoxia exposure affects DNA damage and repair in mouse embryos, we again performed IF for γH2A.X and PARP1 in 8-cell embryos and morula (*Figure 3D and E*) (n=25 embryos per treatment collected from three independent experiments. Statistical test: t-test). We found that treated embryos with reversine and AZ3146 under hypoxia generated lower levels of γH2A.X (*Figure 3E*) compared with the same treatments under normoxia (*Figure 1E*). Whereas PARP1 levels at the 8-cell stage are reduced in reversine-treated embryos (*Figure 3E*), similar to normoxic conditions (*Figure 1E*). Importantly, at the morula stages, the levels of PARP1 are relatively low compared to

morula stage in normoxic conditions (*Figure 1E*). Overall, these data suggest that hypoxia exposure lowers the accumulation of DNA damage between the 8-cell to morula stages and results in a reduction of PARP1 at the morula stage.

## Hypoxia increases the proportion of euploid cells in the EPI of mosaic blastocysts

Despite the high incidence of mosaicism in human pre-implantation embryos, the fate of aneuploid cells in mosaic embryos is incompletely understood. We previously used reversine to generate a mouse model of pre-implantation chromosome mosaicism and found that aneuploid (reversine-treated) cells are eliminated in the EPI of mosaic embryos via apoptosis, starting from the mature blastocyst stage (*Bolton et al., 2016*). Whether this response reflects reversine treatment specifically, or aneuploidy more generally, is not known. Moreover, how hypoxia versus normoxia affects the outcomes is also not clear.

To address these questions, we created aggregation chimeras at the 8-cell stage that contained a 1:1 ratio of AZ3146-treated and control blastomeres (DMSO/AZ3146 chimeras), which is expected to reflect low-grade mosaicism, or of AZ3146-treated and reversine-treated blastomeres (reversine/AZ3146 chimeras), which is expected to reflect medium-grade mosaicism (*Figure 4A*). We cultured these chimeric embryos in normoxia and hypoxia and followed the fate of individual blastomeres by using transgenic mouse lines with the membrane markers mTmG (*Muzumdar et al., 2007*) and E-cadherin (*Christodoulou et al., 2019*). DMSO/AZ3146 and reversine/AZ3146 blastocysts displayed proper lineage allocation, embryo morphology, and cavitation in both normoxia and hypoxia (*Figure 4B and C*). In addition, DMSO/AZ3146 and reversine/AZ3146 embryos had comparable total cell numbers in their blastocysts, and blastocysts grown in hypoxia were again smaller than those grown in normoxia (*Figure 4D and E*). Intriguingly, despite having fewer total cells, DMSO/AZ3146 blastocysts had more EPI cells when they were cultured in hypoxia compared to normoxia (*Figure 4D and E*).

We quantified the proportion of AZ3146 cells in each lineage for each chimera. In DMSO/AZ3146-treated blastocysts, we found that 46.75% of the TE and 42.88% of the EPI originated from AZ3146-treated blastomeres, compared to only 28.57% of the PE (*Figure 4F*). In reversine/AZ3146 chimeras, 63% of the TE and 78.4% of the EPI originated from AZ3146-treated cells, compared to only 40% of the PE (*Figure 4F*). Thus, under normoxia, DMSO cells appear to outcompete AZ3146-treated cells, which in turn outcompete reversine-treated cells in the TE and EPI. AZ3146-treated cells appeared to be at a competitive disadvantage in the PE in both contexts. Culturing DMSO/AZ3146 chimeras in hypoxia increased the representation of AZ3146-treated blastomeres in the TE (49.06%) and PE (50%) but, strikingly, reduced their representation in the EPI (33.3%) (n>7 embryos per treatment collected from three independent experiments. Statistical test: Mann–Whitney U-test) (*Figure 4G*). These altered frequencies reflect a higher number of DMSO-treated cells in the EPI and a lower number in the PE (*Figure 4E*). Hypoxia exposure of reversine/AZ3146 embryos lowered the contribution of AZ3146-treated blastomeres to the TE (55.94%) and EPI (50%), but not to the PE (46.43%) (*Figure 4G*). We found that there was no correlation between the proportion of AZ3146-treated cells in the TE and the EPI in DMSO/AZ3146 and reversine/AZ3146 aggregation chimeras, under hypoxia or normoxia, and that hypoxia seemed to have a greater impact on the proportion of AZ3146 cells in reversine/AZ3146 aggregation chimeras (*Figure 4—figure supplement 1*). Overall, these data suggest that hypoxia has lineage-specific effects on competitions between aneuploid and euploid cells and increases the contribution of euploid cells to the EPI.

Blastomeres in the 4-cell stage embryo display a lineage bias (*Goolam et al., 2016*). We considered that mosaicism generated before the 4-cell stage might influence lineage allocation. To test this possibility, we treated zygotes with reversine or AZ3146 during the first cleavage division. Importantly, reversine treatment at the zygote stage seems to strongly affect the morphology of the blastocysts (*Figure 4—figure supplement 2A*). Consistent with this change in morphology, reversine treatment reduced the number of cells in all lineages, particularly in the TE and PE (*Figure 4—figure supplement 2B*). In contrast, AZ3146 treatment did not affect morphology or cell number in any of the lineages (*Figure 4—figure supplement 2A and B*) (n=~27 embryos per treatment collected from three independent experiments. Statistical test: Mann–Whitney U-test). To evaluate how early generation of aneuploidies in the embryos affects cell competition, we generated DMSO/AZ3146, reversine/DMSO, and reversine/AZ3146 aggregation chimeras at the 2-cell stage and cultured them

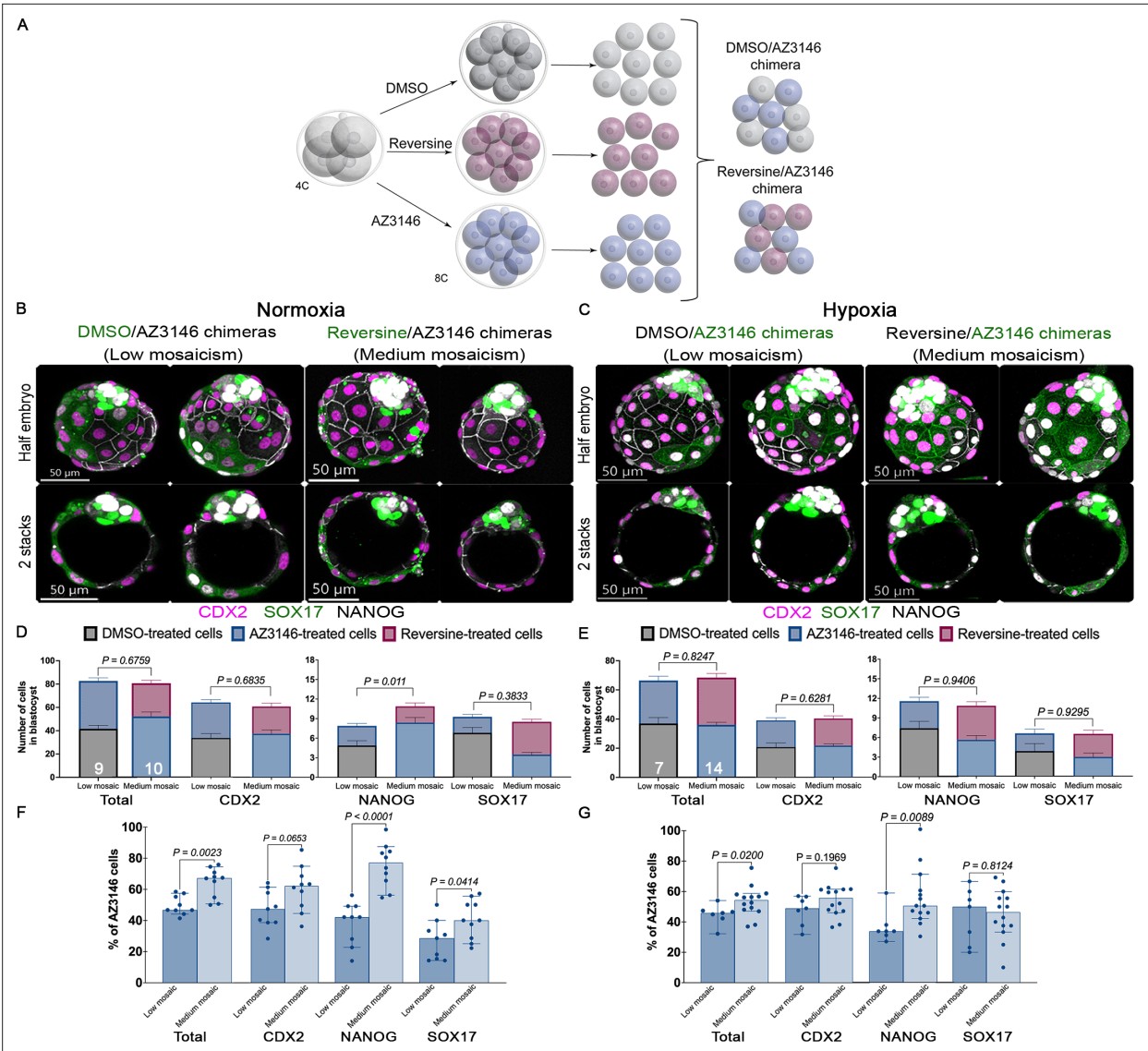

**Figure 4.** Hypoxia affects cell competition between diploid and aneuploid cells during pre-implantation development. (**A**) Graphic representation of the cell competition experiments. Embryos were treated at the 4-cell stage with DMSO or Mps1 inhibitors reversine and AZ3146. After washing, 8-cell stage embryos were disaggregated and re-aggregated to form chimeras containing a 1:1 ratio of DMSO/AZ314-treated blastomeres and reversine/AZ314-treated blastomeres. Following aggregation, chimeras were cultured to the mature blastocyst stage (E4.5) and analyzed for lineage specification. For identification of the treatment, we use transgenic lines with membrane markers. Immunofluorescence for CDX2, NANOG, and SOX17 was performed to test lineage specification and allocation during (**B**) normoxia and (**C**) hypoxia. Lineage allocation quantification was based on the above markers. Importantly, (**D**) in normoxia, both chimeras have the same number of cells in all the lineages except the epiblast (EPI). In addition, AZ3146-treated blastomeres outcompete reversine-treated blastomeres in medium-grade mosaics in the trophectoderm (TE) and the EPI. (**E**) Under hypoxia, both chimeras have a similar number of cells in all the lineages. Yet, AZ3146-blastomeres do not outcompete reversine-treated blastomeres. Quantification of the contribution of AZ3146-treated blastomeres to the chimeras showed that, (**F**) under normoxia, compared with DMSO-treated cells, no preferential allocation of aneuploid cells occurs in the TE. In contrast, AZ3146-treated blastomeres increased their contribution when compared with reversine-treated blastomeres, but only for the EPI and the primitive endoderm (PE). (**G**) During hypoxia, in DMSO/AZ3146 chimeras, no preferential allocation of aneuploid cells occurs in the TE. But preferential allocation of diploid cells to the EPI is observed. Whereas in reversine/AZ3146 chimeras, AZ3146-treated blastomeres contribute similarly to reversine blastomeres to the TE and PE but significantly increase contribution to the EPI. These results indicate that hypoxia favors the survival of reversine-induced aneuploid cells compared to their survival in normoxia (n>7 embryos per treatment collected from three independent experiments. Statistical test: Mann–Whitney U-test, error bars represent s.e.m.).

The online version of this article includes the following source data and figure supplement(s) for figure 4:

**Source data 1.** Quantification of lineage allocation in chimeric embryos.

**Figure supplement 1.** Correlation plot of AZ3146-treated cells in the epiblast and trophectoderm.

**Figure supplement 2.** Aneuploidies generated before the 2-cell stage do not affect lineage specification in mosaic embryos.

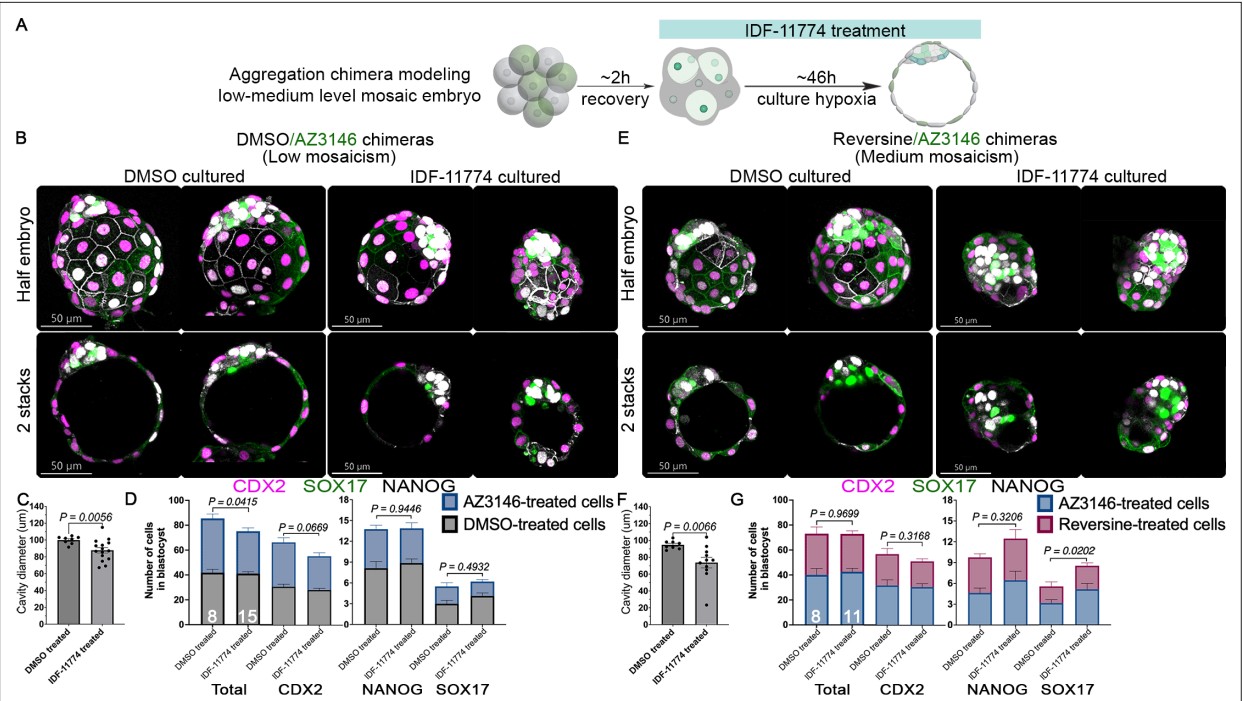

**Figure 5.** HIF1A inhibition increases the proportion of euploid cells in mosaic embryos. (**A**) Schematic of HIF1A inhibition by IDF-11774 in 8-cell stage aggregation chimeras cultured in hypoxia. Immunofluorescence for CDX2, NANOG, and SOX17 was performed to test lineage specification and allocation. (**B**) HIF1A inhibition in low-grade mosaicism does not affect overall morphology but affects (**C**) cavity diameter. In addition, (**D**) lineage allocation quantification reveals a significant reduction of total cell number, as well as a reduction in cell number in the TE but not in the epiblast (EPI) and primitive endoderm (PE). In contrast, (**E**) HIF1A inhibition in medium-grade mosaicism affects morphology and (**F**) cavitation of the mosaic embryos. However, lineage allocation quantification revealed that total cell number is not affected (**G**) in medium-grade mosaicism after IDF11-774 treatment. Nevertheless, HIF1A inhibition appears to increase the cell number of the PE. These results show that HIF1A inhibition in hypoxic conditions differentially affects each type of mosaic embryos (n>8 embryos per treatment collected from three independent experiments. Statistical test: Mann–Whitney U-test, error bars represent s.e.m.).

The online version of this article includes the following figure supplement(s) for figure 5:

**Figure supplement 1.** IDF-11774-mediated inhibition of HIF1A increases the proportion of euploid cells in mosaic embryos.

under normoxia until the blastocyst stage (*Figure 4—figure supplement 2C*). We found that all three 2-cell stage aggregation chimeras developed into blastocysts with proper lineage segregation and the presence of a cavity (*Figure 4—figure supplement 2D*). Interestingly, the reversine-treated cells were extruded from the blastocyst in 54% of the reversine/DMSO chimeras and in 33% of the AZ3146/reversine chimeras, whereas no cell extrusion was observed in the DMSO/AZ3146 chimeras. Quantification of the proportion of AZ3146-treated cells in 2-cell stage-derived chimeras (*Figure 4—figure supplement 2E*) showed similar results as 8-cell stage-derived chimeras (*Figure 4F–G*). In DMSO/AZ3146-treated blastocysts, we found that 49.28% of the TE, 39% of the EPI, and 33.33% of the PE originated from AZ3146-treated blastomeres (*Figure 4—figure supplement 2E*). In reversine/AZ3146 chimeras, 68.24% of the TE, 68.83% of the EPI, and 66.67% of the PE originated from AZ3146-treated cells (*Figure 4—figure supplement 2E*). In DMSO/reversine chimeras, 22.27% of the TE, 26.98% of the PE, and 31.7% of the EPI originated from reversine-treated cells. Overall, these results suggest that aneuploidy generated at different stages similarly affects the proportion of aneuploid cells in the blastocysts.

## HIF1A inhibition reduces the frequency of aneuploid cells in DMSO/AZ3146 mosaic embryos

Our data suggest that hypoxia affects aneuploid–euploid cell competition in mosaic embryos (*Figure 4*). To further assess the effect of HIF1A, and considering most human embryos are cultured under hypoxic conditions, we decided to evaluate the effect of inhibiting HIF1A in mosaic embryos

cultured in hypoxic conditions. We assembled DMSO/AZ3146 and reversine/AZ3146 8-cell stage chimeras, treated them with the HIF1A inhibitor IDF-11774 (*Figure 5A*, *Figure 5—figure supplement 1A and B*), and then assessed lineage allocation in blastocysts. Treatment of DMSO/AZ3146 chimeras with IDF11774 does not affect lineage segregation in the blastocyst but reduces the cavity diameter (*Figure 5B and C*). Additionally, diminish the contribution of AZ3146-treated cells to the TE, but DMSO-treated cells seem to compensate for this loss in the PE and EPI (*Figure 5D*, *Figure 5—figure supplement 1C*). Treating reversine/AZ3146 chimeras with IDF11774 compromised blastocyst morphology and cavitation (*Figure 5E and F*) and considerably increased the number of PE and EPI cells (n>8 embryos per treatment collected from three independent experiments. Statistical test: Mann–Whitney U-test) (*Figure 5G*), without altering the frequency of AZ3146-treated cells (*Figure 5—figure supplement 1D*). Taken together, our data suggest that HIF1A promotes the survival of AZ3146-treated cells in DMSO/AZ3146 chimeras and suggest that inhibiting HIF1A could increase the proportion of karyotypically normal cells in mosaic embryos.

## Discussion

Aneuploidy is a frequent outcome of early embryonic divisions (*Allais and FitzHarris, 2022*; *Currie et al., 2022*; *Palmerola et al., 2022*). Here, we describe complementary strategies for inducing aneuploidy in mouse embryos and reveal how the embryo's response varies depending on the nature of the insult, cell lineage, and environmental context. Our findings support four key conclusions: (1) The Mps1 inhibitor AZ3146 induces aneuploidy while preserving DNA repair capacity and dampening stress responses, compared to the broader-acting inhibitor reversine. (2) Lineage-specific responses to aneuploidy emerge during blastocyst development, consistent with our earlier findings (*Bolton et al., 2016*; *Singla et al., 2020*). (3) HIF1A promotes survival of extra-embryonic lineages (TE and PE) in AZ3146-treated embryos, whereas PARP1 is particularly required in the EPI of both AZ3146- and reversine-treated embryos. (4) Hypoxia enhances DNA repair and influences the competitive dynamics between aneuploid and euploid cells in mosaic embryos.

Reversine-treated embryos upregulate mRNA levels of senescent markers p53 and p21 (*Singla et al., 2020*) and fail to give rise to viable embryos (*Bolton et al., 2016*). Our current results further indicate that reversine downregulates PARP1, consistent with previous findings in glioma cells (*Hirakata et al., 2021*), thereby exacerbating DNA damage and limiting DNA repair capacity. In contrast, AZ3146-treated embryos maintain PARP1 expression and exhibit increased HIF1A activity, which appears to support the survival of extra-embryonic lineages and improve overall developmental potential.

These findings intersect with a broader conversation about embryo culture conditions. It has been proposed that standard culture practices shift from normoxia (20% oxygen) to more physiologic hypoxia (5% oxygen) to better recapitulate the in vivo-like oviductal environment (*Alva et al., 2022*). Indeed, hypoxic culture conditions have been shown to improve the quality of mammalian blastocysts (*Ciray et al., 2009*; *Nguyen et al., 2020*), though the underlying mechanisms remain unclear. Studies in somatic cells suggest that hypoxia enhances DNA repair pathways (*Martí et al., 2021*; *Nakamura et al., 2022*; *Pietrzak et al., 2018*). In line with this, we observed reduced DNA damage in aneuploid embryos cultured under hypoxic conditions, consistent with prior reports of lower γH2A.X levels in mouse blastocysts recovered from the uterus or cultured in low oxygen (*Houghton, 2021*; *Meuter et al., 2014*). Furthermore, recent work in cancer cells indicates that hypoxia may promote HIF1A-PARP1 interactions, enhancing DNA repair (*Martí et al., 2021*; *Nakamura et al., 2022*; *Pietrzak et al., 2018*). Together, these findings highlight the interplay between chromosomal stress, lineage context, and the embryonic microenvironment in shaping cell fate during early development. A deeper mechanistic understanding of how DNA repair is regulated at pre-implantation stages will be crucial for elucidating how embryos tolerate or eliminate aneuploid cells – with important implications for improving embryo culture systems and fertility treatments.

Confined placental mosaicism, in which the placenta is aneuploid while the fetus is euploid, is a common phenomenon in human pregnancies (*McCoy, 2017*). In previous work, we showed that aneuploid TE cells exhibit prolonged cell cycles and signs of senescence, whereas aneuploid ICM cells have an increased frequency of apoptosis in reversine-treated embryos (*Singla et al., 2020*). These findings suggest that abnormal, yet viable TE cells may contribute to confined placental mosaicism. In the present study, we found that in AZ3146-treated embryos cultured under normoxic

conditions, aneuploid cells contributed predominantly to the TE, followed by the EPI, and were especially depleted from the PE. Under hypoxic conditions, however, we observed a shift: aneuploid cells were more frequently eliminated from the EPI and enriched in the PE. These observations suggest that hypoxia modulates lineage-specific cell competition in mosaic embryos, potentially promoting a higher proportion of euploid cells in the EPI relative to the PE and TE. Moreover, we confirmed that the proportions of aneuploid cells in the TE and EPI do not correlate, consistent with the preferential survival of aneuploid cells in the TE and their selective elimination from the ICM. This is in line with clinical data showing low concordance of aneuploidy between TE and ICM in human mosaic embryos (30–40%) (*Dahdouh and Garcia-Velasco, 2021*). Lineage-specific elimination of aneuploid cells by apoptosis has been described in an in vitro model of human post-implantation development using reversine (*Yang et al., 2021*). Our findings mirror this distinction and further suggest that mechanisms of DNA repair – possibly involving PARP1 and HIF1A – may underlie the differential survival of aneuploid cells across lineages. Investigating how these pathways operate during post-implantation development will be crucial for understanding the developmental trajectories of mosaic embryos and the basis of placental mosaicism.

Since low- and medium-grade mosaic human embryos have similar developmental potential to fully euploid embryos (*Capalbo et al., 2021*; *Greco et al., 2015*), understanding the molecular pathways that govern the response to aneuploidy holds important translational promise. Given that the extent and proportion of aneuploid cells in mosaic embryos impact embryo viability (*Capalbo et al., 2021*), strategies that reduce the aneuploid cell burden at the blastocyst stage may enhance the likelihood of a successful pregnancy. In this study, we targeted HIF1A as a potential regulator of aneuploid cell persistence and found that treatment with the HIF1A inhibitor IDF-11774 increased the proportion of euploid cells in mosaic embryos. Taken together, our findings reveal that specific environmental and molecular factors can modulate the composition of mosaic embryos, with potential implications for improving embryo quality and reproductive outcomes.

## Materials and methods
### Pre-implantation embryo culture

Mice were maintained according to national and international guidelines. Four- to six-week-old B6SJLF1/J (RRID:IMSR_JAX:000664) female mice were injected with 7.5 IU PMSG followed by 7.5 IU hCG 48 hr later, to induce superovulation. The females were then mated with B6CBAF1/J (RRID:IMSR_JAX:100011) males or were indicated, with E-cadherin GFP or mTmG transgenic males. Pre-implantation mouse embryos were recovered 24 hr and 40 hr after the hCG injection to obtain zygotes and 2-cell stage embryos, respectively, in M2 medium (Sigma, M7167). We incubate the embryos in KSOM until the 4-cell stage for the aneuploid treatment, around 53 hr post hCG. When recovering zygotes, cumulus cells were removed with 0.3% hyaluronidase (Sigma, H4272) in M2. Embryos were cultured in regular KSOM (Sigma, MR-106) at 37°C under 5% $CO_2$/air (normoxia) or with premixed 5% $CO_2$/5% $O_2$ balance with nitrogen, biologic atmosphere batch (Airgas #Z03NI9022000033) (hypoxia). Normoxia in our study was defined as the standard atmospheric oxygen levels in culture incubators (*Alva et al., 2022*), whereas hypoxia (5% oxygen concentration) was the standard level of oxygen in the IVF clinics and can be used to better model physiologic oxygen (physoxia), which varies from ~1.5% to 8% (*Alva et al., 2022*; *Knudtson et al., 2022*). B6SJLF1/J females and B6CBAF1/J males were obtained from JAX laboratories. Females were received at 3–4 weeks of age and were maintained in the Caltech animal facility, where they were housed with five same-sex littermates on a 12 hr light/12 hr dark cycle with food and water ad libitum. The temperature in the facility was controlled and maintained at 21°C. All experimental procedures involving the use of live animals or their tissues were performed in accordance with the NIH guidelines and approved by the Institutional Animal Care and Use Committee (IACUC) and the Institutional Biosafety Committee at the California Institute of Technology (Caltech, protocol number 1772). Reversine (Sigma, R3904), AZ3146 (Sigma, SML1427), PX-478 (Selleckchem, S7612), IDF-11774 (Selleckchem, S8771), and olaparib (Selleckchem, S1060) were dissolved in DMSO (Sigma, D2650) before use to specific concentration. They were respectively used at the following final concentrations: 0.5 µM, 20 µM, 2 µM, 20 µM, and 10 µM. Control embryos were incubated in the equivalent DMSO concentration. Drugs were dissolved in regular KSOM to the concentration of use. DMSO concentration in media should never pass 0.4%.

## Immunofluorescence

Embryos were fixed in 4% PFA (Thermo Scientific, AA47340) for 20 min at room temperature (RT), followed by three washes with 0.1% Tween-20 (Sigma, P1379) dissolved in PBS (PBST). The embryos were then permeabilized with 0.3%, Triton X-100 (Sigma, X100) in PBS. Washes were then performed three times in PBST before embryos were transferred to blocking solution (3% BSA in PBS) for at least 3 hr at RT. Incubation with primary antibodies was performed in blocking solution overnight at 4°C. The next day, washes were performed three times in PBST before incubation with Alexa Fluor secondary antibodies (Thermo Fisher Scientific, 1:500) in blocking solution for 2 hr. Washes were performed three times before incubation with DAPI (Thermo Fisher Scientific Cat# D1306, RRID:AB_2629482) for 5 min. Washes after DAPI were performed two times in PBST before the final incubation in M2. Embryos were then mounted in M2 micro-drops on 35 mm glass-bottom dish (MATTEK, P35G-1.5-14-C). Confocal imaging was carried out using Leica SP8, 40x objective, 1 μM Z-step. Image files were viewed and analyzed using ImageJ and Imaris 9.9 software.

Primary antibodies used: mouse anti-CDX2 (BioGenex Cat# MU392A, RRID:AB_2923402, 1:500), rabbit anti-NANOG (Abcam Cat# ab80892, RRID:AB_2150114, 1:500), goat anti-SOX17 (R&D Systems Cat# AF1924, RRID:AB_355060, 1:300), rabbit anti-HIF1A (Novusbio Cat# NB100-449SS. 1:300), mouse anti-PARP1 (Proteintech Cat# 13371-1-AP, RRID:AB_2160459. 1:500), and rabbit anti-Phospho-Histone H2AX (R&D Systems Cat# MAB2288, RRID:AB_3657890, 1:500).

## In situ chromosome counting

For the determination of aneuploidy, treated 8-cell stage embryos were synchronized in metaphase by 10 hr treatment with 0.03 μg/mL colcemid (Cayman, 15364) diluted in KSOM, followed by a 1 hr treatment with 10 μM Mg132 (Selleckchem, S2619) in KSOM, and finally, a 1 hr treatment with 5 μM dimethylenastron (MedChemExpress, HY-19944) and 10 μM Mg132 in KSOM. Synchronized embryos were then fixed in 2% PFA for 20 min, permeabilized with PBST for 15 min and blocked 3 hr at RT before incubation with human anti-centromere protein antibody (Antibodies Incorporated Cat# 15-234, RRID:AB_2939058. 1:300) overnight at 4°C. The next day, three washes with PBST were performed before a 2 hr incubation with goat anti-Human secondary antibody, Alexa Fluor 647 (Invitrogen Cat# A-21445, RRID:AB_2535862. 1:400) in blocking solution. Washes were performed three times before 20 min incubation with DAPI (1:500) and Alexa Fluor 488 Phalloidin (Invitrogen Cat# A-12379. 1:300). Washes were then performed three times before clearing overnight with AF1 plus (Citi-fluor, AF1/DAPI-15). Embryos were then mounted in AF1 plus micro-drops on 35 mm glass bottom. Confocal imaging was carried out using Leica SP8, 60x objective, 0.5 μM Z-step. Image files were viewed and analyzed using ImageJ and Imaris 9.9 software.

## Embryo transfers and post-implantation recovery and biopsy

Embryo transfer was performed as described previously (*Bermejo-Alvarez et al., 2014*). CD1 females and vasectomized CD1 males were obtained from Charles River (RRID:IMSR_CRL:022). Reversine and AZ3146-treated blastocysts were transferred into 2.5 days pseudopregnant CD1 females. Around 16 AZ3146-treated embryos were transferred in the right uterine horn, whereas around 16 reversine-treated embryos were transferred in the left uterine horn of the same female as control. To evaluate implantation and embryo survival potency, uterine horns were recovered 5 days after surgery. Deciduas were dissected and post-implantation embryos at stage E9.5 were recovered into PBST on ice. Embryos were fixed in 4% PFA at RT for 1 hr, followed by washes through PBST before imaging. Images were taken using an Olympus LS stereo microscope with a 10x objective. Uterine transfers were performed in accordance with the NIH guidelines and approved by the Institutional Animal Care and Use Committee (IACUC) and the Institutional Biosafety Committee at the California Institute of Technology (Caltech).

## qRT-PCR

Around 14–17 morulas and blastocysts were collected for quantitative reverse transcriptase polymerase chain reaction (qRT-PCR). Total RNA from morulas was obtained using the NucleoSpin RNA Plus XS kit (Takara, 740990.10). Whereas total RNA from blastocyst was extracted using the Arcturus PicoPure RNA Isolation Kit (Thermo Fisher, KIT0204). qRT-PCR was performed using the *Power* SYBR Green RNA-to-CT 1-Step Kit (Applied Biosystems, 4389986) in a StepOne Plus Real-time PCR machine

(Applied Biosystems). The following program was used: 30 min 48°C (reverse transcription) followed by 10 min 95°C followed by 45 cycles of 15 s 95°C (denaturing) and 1 min 60°C (annealing and extension). The ddCT method was used to determine relative levels of mRNA expression, with *Ppia* as an endogenous control (*Gu et al., 2014*). Primers were obtained from IDT.

| Name | Sequence |
|------|----------|
| *Trp53*_Fw | GTCACAGCACATGACGGAGG |
| *Trp53*_Rv | TCTTCCAGATGCTCGGGATAC |
| *Hif1a*_Fw | CCTGCACTGAATCAAGAGGTTGC |
| *Hif1a*_Rv | CCATCAGAAGGACTTGCTGGCT |
| *Ppia*_Fw | GAGCTCTGAGCACTGGAGAGA |
| *Ppia*_Rv | CCACCCTGGCACATGAAT |

## Generation of chimeric embryos

Chimeric embryos were generated following a previously published protocol (*Eakin and Hadjantonakis, 2006*). Briefly, wild-type, E-cadherin GFP (RRID:IMSR_JAX:016933) or mTmG (RRID:IMSR_JAX:007576) 8-cell stage embryos were transferred to micro-drops of M2 after aneuploid treatment. Zona pellucida was removed by treatment with acidic Tyrode's solution (Sigma, T1788). The embryos were then incubated in Ca2+/Mg2+-free M2 (made in house) for 5 min and then disaggregated into individual blastomeres by gentle mouth pipetting. The chimeric embryos were formed by aggregating AZ3146-treated, reversine-treated or DMSO-treated cells in a 1:1 proportion, 4 cells from one treatment and the other 4 cells with another treatment. Low mosaicism DMSO/AZ3146-treated blastomeres (1:1), whereas medium mosaicism reversine/AZ3146-treated blastomeres (1:1). Culture of the chimeras was performed under normoxic and hypoxia conditions in KSOM for 48 hr to reach blastocyst stage.

## Statistical analysis

The statistical tests used are indicated in the corresponding figure legends. Calculations were carried out in Microsoft Excel and data analysis and visualization in Prism 9 software. All graphs show mean values, error bars: s.e.m.

## Acknowledgements

This work was supported by MZG' National Institutes of Health R01 (R01HD101489) grant. ESV is supported by a Pew Latin America fellowship. We thank the Life Science Editors and the Life Science Editors Foundation for invaluable comments and suggestions on the manuscript, and Ariane Helou for copy editing. Data analysis was performed in the Biological Imaging Facility, with the support of the Caltech Beckman Institute and the Arnold and Mabel Beckman Foundation.

## Additional information

### Funding

| Funder | Grant reference number | Author |
|--------|------------------------|--------|
| National Institutes of Health | R01HD101489 | Magdalena Zernicka-Goetz |
| Pew Charitable Trusts | Pew Latin America fellowship | Estefania Sanchez-Vasquez |

The funders had no role in study design, data collection and interpretation, or the decision to submit the work for publication.

## Author contributions
Estefania Sanchez-Vasquez, Conceptualization, Formal analysis, Validation, Investigation, Visualization, Methodology, Writing – original draft, Writing – review and editing; Marianne E Bronner, Supervision, Writing – review and editing; Magdalena Zernicka-Goetz, Conceptualization, Funding acquisition, Investigation, Visualization, Methodology, Writing – review and editing

## Author ORCIDs
Estefania Sanchez-Vasquez ⓘ https://orcid.org/0000-0002-6585-8548
Marianne E Bronner ⓘ https://orcid.org/0000-0003-4274-1862
Magdalena Zernicka-Goetz ⓘ https://orcid.org/0000-0002-7004-2471

## Ethics
We conducted the experiments in this project according to the highest ethical standards. We followed published NIH guidelines, and our protocols and number of animals used were previously approved by the Institutional Animal Care and Use Committee (IACUC) as well as the Institutional Biosafety Committee at the California Institute of Technology (Caltech, protocol number 1772).

Reviewer #1 (Public review): https://doi.org/10.7554/eLife.101912.4.sa1
Author response https://doi.org/10.7554/eLife.101912.4.sa2

## Additional files

### Supplementary files
MDAR checklist

### Data availability
Source data files have been provided for all figures.

The following previously published dataset was used:

| Author(s) | Year | Dataset title | Dataset URL | Database and Identifier |
|---|---|---|---|---|
| Deng Q, Ramsköld D, Reinius B, Sandberg R | 2014 | Single-cell RNA-Seq reveals dynamic, random monoallelic gene expression in mammalian cells | https://www.ncbi.nlm.nih.gov/geo/query/acc.cgi?acc=GSE45719 | NCBI Gene Expression Omnibus, GSE45719 |

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
