## [Editor Report · eLife Assessment]

Sanchez-Vasquez et al establish an innovative approach to induce aneuploidy in preimplantation embryos. This **important** study extends the author's previous publications evaluating the consequences of aneuploidy in the mammalian embryo. In this work, the authors investigate the developmental potential of aneuploid embryos and characterize changes in gene expression profiles under normoxic and hypoxic culture conditions. Using a **solid** methodology they identify sensitivity to Hif1alpha loss in aneuploid embryos, and in further **convincing** experiments they assess how levels of DNA damage and DNA repair are altered under hypoxic and normoxic conditions.

---

## [Referee Report · Reviewer #1 (Public review)]

Summary:

This paper developed a model of chromosome mosaicism by using a new aneuploidy-inducing drug (AZ3146), and compared this to their previous work where they used reversine, to demonstrate the fate of aneuploid cells during murine preimplantation embryo development. They found that AZ3146 acts similarly to reversine in inducing aneuploidy in embryos, but interestingly showed that the developmental potential of embryos is higher in AZ3146-treated vs. reversine-treated embryos. This difference was associated with changes in HIF1A, p53 gene regulation, DNA damage, and fate of euploid and aneuploid cells when embryos were cultured in a hypoxic environment.

Strengths:

In the current study, the authors investigate the fate of aneuploid cells in the preimplantation murine embryo using a specific aneuploidy-inducing compound to generate embryos that were chimeras of euploid and aneuploid cells. The strength of the work is that they investigate the developmental potential and changes in gene expression profiles under normoxic and hypoxic culture conditions. Further, they also assessed how levels of DNA damage and DNA repair are altered in these culture conditions. They also assessed the allocation of aneuploid cells to the divergent cell lineages of the blastocyst stage embryo.

---

## [Author Response]

The following is the authors’ response to the previous reviews

**Reviewer #1(Public review):**

We deeply appreciate the reviewer comments on our manuscript. Following up the revisions, our manuscript has been improved thanks to their insightful remarks. We have proceeded with all the required changes.

Weaknesses:The authors have still not addressed the inconsistent/missing description for sample size, the appropriate number of * for each figure panel, and the statistical tests used.

Description of sample size, specific *P* value and statistical test used has been added it both in the main text, figures and figure legends.

The authors assign 5% oxygen as hypoxia. This is not the case as the in vivo environment is close to this value. 5% is normoxia. Clinical IVF/embryo culture occurs at 5% O2. Please adjust your narrative around this.

We define in our manuscript “normoxia” as the standard atmospheric oxygen levels in tissue culture incubators, which range from about 20–21% oxygen. Our definition of hypoxia is 5% concentration of oxygen, taking into consideration the standard levels of oxygen in the IVF clinics. Physiological oxygen in mouse varies from ~1.5% to 8% (Alva et al 2022). Considering that these levels of oxygen are the standard levels in tissue culture practices, a paragraph has been added to the discussion and materials and methods for further clarification

**Reviewer #2 (Public review):**
Weakness:Given that this is a study on the induction of aneuploidy, it would be meaningful to assess aneuploidy immediately after induction, and then again before implantation. This is also applicable to the competition experiments on page 7/8. What is shown is the competitiveness of treated cells. Because the publication centers around aneuploidy, inclusion of such data in the main figure at all relevant points would strengthen it. There is some evaluation of karyotypes only in the supplemental - why? Would be good not to rely on a single assay that the authors appear to not give much importance.

This is an excellent point. However, due to the stochasticity of the arising of aneuploidies when embryos are treated with AZ3146 and reversine (Bolton et al 2016), every treatment is likely to generate different levels of aneuploidy. Due to this, and to the technical limitations of generating single-cell genomic DNA sequencing at the blastocyst stage, we were unable to determine the karyotype of all cells after different conditions. Nevertheless, Regin et al 2024 (eLife) showed similar results on the overall transcriptome changes of different dosages of aneuploidy: high dosage embryos overexpress p53, like reversine-treated embryos; meanwhile, low dosage embryos overexpress the hypoxic pathway, including HIF1A, similar to embryos treated with AZ3146.

**Reviewer #1 (Recommendations for the authors):**
Corrections required before final publishing:Please ensure that the number of asterisks is in alignment with standard convention (* <0.05; ** <0.01; *** <0.001; **** <0.0001). If you want to describe an exact P -vale it should be presented as P = 0.0004. line 108 *** is <0.0004. line 263 * P<0.0044Same issue appears in lines 697, 711, 722, 753, 685

Specific values have been added in the figures and modified in the text.

Line 199: "...viable E9.5 embryos" missing "Figure S1D"

Modified in manuscript

Line 120: "...decidua" please add "Figure S1C"

Modified in manuscript

Line 126-127: Please add a description for the results (morula) in Fig 1D, e.g., It appears that YH2Ax persists from 8-cell to morula when treated with Reversine but not AZ3146"

At the morula stage, the levels of γH2A.X in reversine- and AZ3146-treated embryos are similar (Fig. 1E). However, at the blastocyst stage, high levels of γH2A.X are maintained in reversine-treated embryos and reduced in AZ3146-treated embryos, suggesting some level of DNA repair between the morula-to-blastocyst stages (Fig. S2A). In contrast, in hypoxia, the levels of γH2A.X are low in the three treatments at the morula stages, suggesting that DNA repair can be enhanced under hypoxic conditions. Similar results have been reported in somatic cells (Marti et al., 2021; Pietrzak et al., 2018).

Line 213: PARP1 levels were also similar under all conditions; but Fig3E, top right shows PARP1 was significantly lower with Reversine treatment; also please correct me if i am wrong, but does the phrase "all conditions" cross reference yH2AX and PARP1 between Fig 3 and Fig 1 to show the impact of hypoxia? Because from my understanding Fig 1 was done in 20% oxygen, but Fig 3 was done in 5% oxygen – hypoxia.

This is correct. Modification in the manuscript has been performed for clarification

Line 264: extra forward dash? "Reversine/AZ3146/ aggregation"

Modified in manuscript

Line 644: you don't have a control for IDF treatment, so how did you differentiate between impact of aneuploid drugs vs IDF treatment alone? Would the impact observed be due to compounding effect of aneuploidy drugs + IDF?

This is a great observation. We previously demonstrated that IDF-1174 treatments in embryos do not affect pre-implantation development (Fig. S3).

Line 681: change their behaviour is a vague statement. Be specific.

Modified in manuscript

Line 676 missing bracket "E"

Modified in manuscript

Line 680: "...significantly on" should be "for"

Modified in manuscript

Line 682-685: "...hypoxia favours the survival of reversine-induced aneuploid cells." does it? the statement before this says in Rev/AZ chimeras, AZ blastomeres contribute similarly to reversine-blastomeres to the TE and PE but significantly increase contributions to the EPI.Wouldn't this mean hypoxia favours survival of AZ aneuploid cells in EPI?

In normoxic conditions, AZ3146 treated cells in Rev/AZ chimeras contributed mostly to the EPI and TE but not PE. In contrast, in normoxic conditions, Rev-treated cells contributed similarly to all the lineages. This result seems to be due to a better survival of Rev-treated cells under normoxic conditions (Fig. 4D-E)

Line 720: (b) shows blastocyst staining from what group? DMSO? Rev/AZ? Or are the 3 blastocysts shown here, 3 separate examples of Reversine-treated blastocysts? Would require labelling Fig S2B, and adding a short description in the corresponding figure legend

Figure (B) shows the expression pattern of PARP1 at the blastocyst stage. Modified in manuscript

Figure 2, Figure S3 and Figure S6: were these experiments performed at 5% or 20% O2, please add detail.

Modified in manuscript

**Reviewer #2 (Recommendations for the authors):**
Lines 45-46 understanding of reduction of aneuploidy should mention/discuss the paper of attrition/selection, of the kind by the Brivanlou lab for instance, or others. As well as allocation to specific lineages, including the authors' work.

A section in the discussion has been added in response to this recommendation. Comparison between models is debatable.

The response does not clarify whether other papers were cited instead, or the authors own work that has shown preferential allocation to TE.